# Growth and Fatty Acid Composition of Black Soldier Fly *Hermetia illucens* (Diptera: Stratiomyidae) Larvae Are Influenced by Dietary Fat Sources and Levels

**DOI:** 10.3390/ani12040486

**Published:** 2022-02-16

**Authors:** Xiangce Li, Yewei Dong, Qiuxuan Sun, Xiaohong Tan, Cuihong You, Yanhua Huang, Meng Zhou

**Affiliations:** Key Laboratory of Aquatic Animal Diseases and Waterfowl Breeding, Provincial Water Environment and Aquatic Products Security Engineering Technology Research Center, CInnovative Institute of Animal Healthy Breeding, Guangzhou Guangdong College of Animal Sciences and Technology, Zhongkai University of Agriculture and Engineering, Guangzhou 510225, China; xiangceli@163.com (X.L.); sunshinewonder@163.com (Y.D.); sun990921@163.com (Q.S.); tanxiaohong@zhku.edu.cn (X.T.); youcuihong@zhku.edu.cn (C.Y.)

**Keywords:** black soldier fly, dietary fat, growth, fatty acid composition

## Abstract

**Simple Summary:**

The black soldier fly *Hermetia illucens* is a potentially promising feed or food source due to its valuable nutritional composition. However, its fat content and fatty acid composition significantly vary with the rearing substrates, making it a flexible resource in fat quantity and quality. Recent research has attempted to manipulate the fat content and fatty acid composition of the insect with modulated feed formulation. Nevertheless, the results of most investigations are not comparable, since the rearing substrates used are various organic wastes with complex substrates. Therefore, it is necessary to perform a quantitative and accurate assessment of dietary fat on larval yield and fatty acid composition. In this study, the influence of two supplemental levels of six dietary fat sources on growth and nutrient composition was evaluated, especially fatty acid composition of black solider fly *Hermetia illucens* larvae. Additionally, the relationships between dietary fat and larval growth and fatty acid composition were quantitatively determined. Our work deepens the understanding of the fat/fatty acid needs of this insect and therefore enlightens the purposive culture of insects for appropriate fat supply.

**Abstract:**

A 16-day rearing trial was performed to investigate the influence of two supplemental levels (5% and 10%) of six dietary fat sources (linseed oil, peanut oil, coconut oil, soybean oil, lard oil and fish oil) on the growth, development and nutrient composition of black solider fly larvae. Our results demonstrated that the pre-pupa rate of larvae was linearly influenced by dietary C18:0, C18:3n-3 and C18:2n-6 content (pre-pupa rate = 0.927 × C18:0 content + 0.301 × C18:3n-3 content-0.258 × C18:2n-6 content *p* < 0.001)), while final body weight was linearly influenced by that of C16:0 (final body weight = 0.758 × C16:0 content, *p* = 0.004). Larval nutrient composition was significantly affected by dietary fat sources and levels, with crude protein, fat and ash content of larvae varying between 52.0 and 57.5, 15.0 and 23.8, and 5.6 and 7.2% dry matter. A higher level of C12:0 (17.4–28.5%), C14:0 (3.9–8.0%) and C16:1n-9 (1.3–4.3%) was determined in larvae fed the diets containing little of them. In comparison, C16:0, C18:1n-9, C18:2n-6 and C18:3n-3 proportions in larvae were linearly related with those in diets, with the slope of the linear equations varying from 0.39 to 0.60. It can be concluded that sufficient C16:0, C18:0 and C18:3n-3 supply is beneficial for larvae growth. Larvae could produce and retain C12:0, C14:0, and C16:1n-9 in vivo, but C16:0, C18:1n-9, C18:2n-6 and C18:3n-3 could only be partly incorporated from diets and the process may be enhanced by a higher amount of dietary fat. Based on the above observation, an accurately calculated amount of black soldier fly larvae could be formulated into aquafeed as the main source of saturated fatty acids and partial source of mono-unsaturated and poly-unsaturated fatty acids to save fish oil.

## 1. Introduction

Research on the exploration of new feed resources for animal feed has been pushed forward due to the increasing demand of animal-derived products’ consumption worldwide expected by 2050 [1]. Insects are quite attractive as an innovative feed source because of their promising nutritive composition and sustainability [2]. The black soldier fly *Hermetia illucens* belongs to the Diptera: Stratiomyidae, with the characteristics of fast reproduction, large biomass, and easy feeding. Black soldier flies are rich in protein, lipids and other active substances, such as chitin, minerals, antimicrobial peptides, and lauric acid, considered one of the best potential animal feed resources [3]. More and more studies revealed that the addition of black solider fly larvae in feed has no negative effects on the growth and health status of various animals, including swine [4], poultry [4,5,6,7], and fish [8,9,10]. In Europe, the US, Canada and China, black soldier fly larvae application was first legally permitted in aquaculture, which might be followed by poultry and livestock culture [11]. Since the developing aquaculture industry has a growing dependency on fish meal and fish oil [12], the black soldier fly larvae could be a sustainable alternative source of both protein and fat in aquatic feeds.

Compared with other insects, the black soldier fly commonly contains a higher amount of fat (up to approximately 40%) and is rich in saturated fatty acids (SFA), especially palmitic acid (C16:0) and lauric acid (C12:0), the latter of which is known for its antimicrobial activity against Gram positive bacteria [13,14,15]. It also has higher contents of oleic acid (C18:1n-9), palmitic acid (C16:0) and linoleic acid (C18:2n-6) in its body compared to other insects [14]. However, the fat content and fatty acid composition of the larvae dramatically vary (ether extracts from 5% to 40% dry matter) with the rearing substrates [16,17,18,19,20]. Therefore, some researchers attempted to produce the larvae with specific fatty acid profiles by adjusting their diet formula [21,22,23,24,25]. For example, larvae fed with fish offal and algae was found to incorporate significant amounts of eicosapentaenoic acid (C20:5n-3) and docosahexaenoic acid (C22:6n-3) [21], and larvae fed with 11 different diets composed of mussels, bread, fish and food waste contained mainly SFA and mono-unsaturated fatty acids (MUFA), and the incorporation of n-3 fatty acids from the diet was limited [17]. However, the above results are incomparable, since the experimental diets are various organic manures with complex substrates. Little is known about how the dietary fats affect the larval nutrient deposition, especially larval fat level and fatty acid composition, as well as which fatty acids are produced by the larvae itself, and which or how much of them are produced by diets. Therefore, a quantitative and accurate assessment of dietary fat on larvae output and fatty acid composition is necessary.

In this study, two supplemental levels (5% and 10%) of six fat sources with different fatty acid composition were formulated into diets to rear the newly hatched black soldier fly larvae for 16 days. The aim was to determine (1) the nutrient composition of black soldier fly larvae and (2) the correlation between the dietary fatty acid compositions and the larval fatty acid compositions, as well as to (3) evaluate if there was a correlation between the larval yield and the dietary fatty acid compositions. The results can deepen the understanding of the fat/fatty acid needs of this insect and be beneficial in manipulating its fatty acid profile for specific applications.

## 2. Materials and Methods

### 2.1. Preparation of Experimental Diets

Six fat sources used in the study were linseed oil, peanut oil, coconut oil, soybean oil, lard oil and fish oil. All the vegetable oil was food-grade cold-pressed oil, and animal oil was primary refined oil. They were provided by China Oil and Foodstuffs Corporation.

Twelve experimental diets were formulated by mixing six fat sources with soybean meal, with a ratio of 95:5 or 90:10. Ingredients were fully mixed with distilled water (moisture content of the mixture was 70 ± 2%) before being fed to the larvae. The formulation and nutrient composition are presented in Table 1. The fatty acid composition of the experimental diets is presented in Table 2.

### 2.2. Rearing Trial of Black Soldier Fly Larvae

The newly hatched black soldier fly larvae provided by the Institute of Animal Science, Guangdong Academy of Agriculture Sciences (Guangzhou, China) were raised on a diet composed of milled soybean meal (50%) and distilled water (50%) for 3 days (average weight 2.16 ± 0.00 mg) before being introduced to the experimental diets. The experimental diets were maintained in the plastic cuboid boxes (17.5 × 8.0 × 6.5 cm/box) covered with mesh nettings, with four boxes per diet, 200 larvae in each box. Temperature, air humidity and photoperiod during the rearing period were maintained at 28 ± 1 °C, 70–80% relative humidity and 12:12 light:darkness in a lab room (6 m^2^) equipped with an air conditioner and a humidifier. The rearing trial lasted for 16 days from 3-day-old to 19-day-old larvae, during which the emerged pre-pupa in each box was counted for pre-pupa rate calculation. Before harvesting, the larvae individuals were separated from the rearing residual, cleaned by distilled water and air-dried for sampling.

### 2.3. Sampling

Then, 20 individuals from each group were randomly selected, washed in phosphate buffer saline and wiped dry by filter paper for body size and weight measurement, by using vernier caliper (JING, SIDA Rang: 150 mm) and electronic balance (Sartorius, BSA224S-CW). About 100 larvae individuals from each box were collected and stored at −40 °C for proximate nutrients and fatty acid composition analysis.

### 2.4. Proximate Nutrients Analysis

Moisture content was determined by drying samples in an oven at 105 °C for 24 h; nitrogen content was determined using the Kjeldahl method (ISO 5983–1, 2005) and converted to crude protein content by multiplication with factor 6.25. Crude fat was analyzed by the Soxtec extractor method with the Soxtec System (Tecator, Hoganas, Sweden). Crude ash content was analyzed by drying samples at 550 °C by using a muffle furnace. Amino acids (FAA) were analyzed by an automatic amino acid analyzer (Hitachi, L-8900, Tokyo, Japan), gross energy was determined by microbomb calorimeter (Philipson, Gentry Instruments Ins, Aiken, SC, USA), calcium content was determined by the method of EDTA complexometric titration, phosphorus content was determined by method of ammonium vanadate-molybdate, chlorine and sodium content were determined using atomic absorption Spectrum (240FS AA, Agilent, CA, USA)

### 2.5. Fatty Acid Composition Analysis

Frozen samples were freeze-dried to constant weight, and lipid were extracted from these samples using the direct methylation method as described by [26]. Fatty acid composition of the samples was determined after methylation with 14% BF3 methyl esters (FAME) by gas-liquid chromatography on a 7890A instrument (Agilent) using a DB-23 column (30 m × 0.25 mm × 0.20 μm; Agilent) under the following conditions: Detector: hydrogen flame detector (FID); Injection Volume: 1 µL; Inlet heater: 260 °C; Split ratio: 35:1; Detector heater: 280 °C; H_2_ flow: 40 mL/min; Air flow: 400 mL/min; Purge flow: 25 mL/min; Gas: 99.999% pure; Air pressure: 0.4 Mpa; N_2_ pressure: 0.5–0.8 Mpa; H_2_ pressure: 0.3–0.4 Mpa. The fatty acid composition was expressed as percentage of identified fatty acids. The percentages of SFA, MUFA and PUFA in diets or larvae were calculated by adding the total percentage of the identified saturated fatty acids, mono-unsaturated fatty acids or poly-unsaturated fatty acids.

### 2.6. Statistical Analysis

Statistical analyses were performed by using IBM SPSS 23.0 software. Normality and homoscedasticity assumptions were confirmed prior to any statistical analysis. The data which did not conform to the normal distribution were transformed before analysis. One-way ANOVA was conducted to establish significant variance in the growth parameters (pre-pupa rate, final body weight, final body size), as well as proximate nutrients (moisture, crude protein, crude fat and crude ash) of larvae, followed by Tukey’s test for post hoc testing, so as to compare the significance between groups. *p* < 0.05 indicated a significant difference between the values compared.

The multiple stepwise regression analysis was performed to estimate the influence of various fatty acid contents in diets on larval growth and development, with the content of individual fatty acid in diets (% dry weight) as the independent variable (x_1_, x_2_, ……, x_n_), and larval pre-pupa rate (%) or final body weight (mg) as dependent variable (y), respectively. *p*-Values <  0.05 mean a significant influence of the independent variable on the dependent variable. Before that, the residual error distribution test and multicollinearity analysis were conducted to check for outliers and avoid multi-pertinence of factors. 

With the proportion of individual fatty acid (% fat) as independent variable (x), and the proportion of larval fatty acid (% fat) as dependent variable (y), one variable linear regression analysis was performed to evaluate the effects of fatty acid composition in diets on those of larvae. In addition, initially scatter plots with regression lines of each factor were first generated, and the best-fit regression model was then selected based on the adjusted R^2^ value. *p*-Values <  0.05 indicate a significant influence of the independent variable on the dependent variable.

## 3. Results

### 3.1. Growth 

Growth parameters of black soldier fly larvae fed with different dietary fat levels and sources are listed in Table 3.

It was found that the larvae accepted all diets well, while the pre-pupa emerged asynchronously in different diets. At the end of the feeding trial, no pre-pupa was observed in the linseed oil, peanut oil and coconut oil groups at 5% supplemental fat level, and only a few pre-pupa (with the pre-pupa rate of 1.1% and 1.2%) in the soybean oil and fish oil groups, which were significantly lower than that of the lard oil group (7.7%). The pre-pupa rate significantly increased as supplemental fat increased from 5% to 10% in the linseed oil, coconut oil and lard oil groups. At the 10% level, the highest pre-pupa rate was achieved in the lard oil group, followed by that of linseed oil, fish oil and coconut oil, and lowest in the soybean oil group. No pre-pupa was observed in the peanut oil group during the whole rearing trial at both fat levels.

Regarding both supplemental fat levels, the final body weight of the larvae in the fish and lard oil groups was significantly higher than that in the other groups, followed by that in coconut and soybean oil groups, and lowest in the linseed oil and peanut oil groups (Figure 1 and Table 3). With the supplemental fat increasing from 5% to 10%, final body weight significantly increased in the linseed oil, peanut oil, coconut oil and fish oil groups by 20.0%, 68.5%, 11.4% and 29.5%, respectively. The final body size of the larvae exhibited a similar change tendency with that of the final body weight. Nonetheless, the variation range was smaller. A significant increase in the final body size in peanut oil, coconut oil and fish oil groups was observed as the supplemental fat increased from 5% to 10%. At the 10% level, the highest final body weight and size were achieved in the fish oil group, and lowest in the linseed oil group.

The multiple stepwise regression analysis on pre-pupa rate and final body weight suggested that the content of C18:0, C18:3n-3 and C18:2n-6 in diet was the major factor affecting the pre-pupa rate, with the coefficient of determination R^2^ of the regression mode to be 0.899 (*p* < 0.05), showing significant linear relationship and higher goodness-of-fit. The content of 16:0 in diet was the major factor affecting larval final body weight, and the correlation coefficient for optimal regression equation was 0.531 (*p* < 0.05). According to the above regression models, C18:0, C18:3n-3 and C18:2n-6 contents in diets explained 92.7%, 30.1% and −25.8% of the variation in larval pre-pupa rates, and 16:0 content in diets explained 53.1% of the variation in larval final body weight (Table 4).

### 3.2. The Nutrient Composition of Larvae

The nutrient composition of black solider fly larvae was significantly influenced by both fat sources and levels in diets (Figure 2). The moisture, crude protein and crude ash content of the larvae fed with each fat source decreased with the supplemental fat increasing from 5% to 10%. On the contrary, the crude fat of the larvae significantly increased as supplemental fat increased.

At 5% supplemental fat level, significantly lower moisture content of larvae in the linseed oil and soybean oil groups was observed, while crude protein content of larvae in linseed oil and peanut oil groups was the highest, followed by that of coconut oil, lard oil, fish oil and soybean oil groups. Similar results were observed at 10% fat level. Specifically, the crude protein of larvae reached the highest level in the linseed oil and peanut oil groups, and the lowest in the coconut oil group. At both dietary fat levels, the highest and lowest value of larval crude fat was observed in the lard oil and peanut oil groups. It altered from 18.6% to 24.5% at the 10% supplemental fat level and 15.0% to 20.9% at the 5% supplemental fat level. The crude ash content was highest in the peanut oil group, followed by that of other groups, with significant differences at both dietary fat levels.

### 3.3. Fatty Acid Composition

The fatty acid composition of the experimental diets was quite different. At both supplemental fat levels, C18:3n-3 (46.5% and 50.0%) and C18:1n-9 (19.5% and 18.7%) in the linseed oil diet, 18:1n-9 (43.6% and 45.6%) in the peanut oil diet, C12:0 (38.6% and 42.7%) in the coconut oil diet, C18:2n-6 (49.8%) in the soybean oil diet, and C18:1n-9 (37.2% and 38.5%) and C16:0 (24.9% and 25.9%) in the lard oil diet were the dominant fatty acids (accounting for more than 40% of the total fatty acids), respectively. The fish oil diet was the only diet containing C20:5n-3 (7.4% and 8.3%) and C22:6n-3 (10.7% and 12.1%) (Table 2). As a result, proportions of saturated fatty acid (11.7–84.80%), mono-unsaturated fatty acid (7.8–45.0%) and poly-unsaturated fatty acid (7.4–69.3%) largely varied between diets.

Compared with that of the initial larvae, the fatty acid composition of the final larvae changed greatly. C16:0 (12.9%), C18:0 (13.5%), C18:1n-9 (29.5%) and C18:2n-6 (33.8%) were the dominant fatty acids in the initial larvae, while C12:0 (17.4–44.9%), C14:0 (3.9–15.2%), C16:0 (10.3–21.7%), C16:1n-9 (1.3–6.7%), C18:1n-9 (11.9–35.7%) and C18:2n-6 (5.6–26.1%) were the dominant fatty acids in the harvested larvae (Table 5). Much higher proportions of C12:0, C14:0, and C16:1n-9 in larvae over those in diets between groups were observed. The variation of the proportions of saturated fatty acid (43.6–77.3%), mono-unsaturated fatty acid (16.5–37.9%) and poly-unsaturated fatty acid (6.2–37.1%) in larvae was smaller between groups compared to diets.

One variable linear regression analysis was conducted to evaluate the influence of dietary fatty acid composition on the diets of larvae (Figure 3). It was revealed that the proportions of C16:0, C18:1n-9, C18:2n-6 and C18:3n-3 in larvae was positively influenced by those in diets at both supplemental fat levels, with the slopes of linear regression equations varying from 0.39 to 0.60. The slopes of linear regression equations for the four fatty acids in the 10% fat level groups were a little larger than those in the 5% level groups: 0.52 (10%) vs. 0.45 (5%) for 16:0, 0.60 (10%) vs. 0.47 (5%) for 18:1n-9, 0.50 (10%) vs. 0.43 (5%) for 18:2n-6, and 0.46 (10%) vs. 0.39 (5%) for 18:3n-3. The results suggested that fatty acid incorporation may be enhanced when diets contain more fat.

## 4. Discussion

### 4.1. Growth of Larvae

The insects, though they have evolved to survive on a nutritionally imbalanced diet, require dietary lipids for optimal growth, especially those during the special physiological periods (metamorphosis, diapause or starvation, etc.) [27,28]. To date, little data about the dietary fat requirements of black soldier fly are available, except a recent study revealing an increase in larval weight gain with dietary fat level increasing from 3% to 12% [29]. In the present study, the final body weight and size of larvae fed with linseed oil, peanut oil, coconut oil and fish oil significantly increased when the supplemental fat level increased from 5% to 10%. However, they remained unchanged in larvae fed with soybean oil and lard oil. The results implied that, not only the energy level of diets, but some fatty acids may act as crucial nutrients for larvae growth, and dietary fat sources containing such fatty acids were required until appropriate levels. Additionally, C16:0 in this study was statistically identified as a specific fatty acid, whose content positively affected the final body weight of the larvae. Previous studies demonstrated that many insect species preferred to convert other fatty acids into C16:0 for the next lipogenesis [30], and C16:0 was one of the substrates for pheromone biosynthesis [31]. Therefore, we speculated that C16:0 is existing in the fatty acid pool, into which other dietary fatty acids are converted for further lipid synthesis, like most insects [30]. In *Drosophila*, its pupa mitochondria preferentially oxidized medium-chain (C12) rather than long-chain (C16) acyl-CoA, which was preferentially used for anabolic metabolism [32]. However, excessive C16:0 in diets may induce an imbalance of glucose and lipid metabolism [33], oxidative stress, and inflammation, decreasing the lifespan of *Drosophila* [34]. Under the consideration of the specific genetic properties of the black soldier fly, which are different from other fly species [35], more investigations on the roles of 16:0 participating in fat metabolism for the species are needed.

Interestingly, fish oil inclusion in diets rising from 5% to 10% induced a great increase of larval final body weight and size. This was even significantly higher than that of the 10% lard oil group. Notably, C16:0 content in fish oil was no more than that in lard oil, but fish oil was the only oil source that contained 20:5n-3 (EPA) and 22:6n-3 (DHA). Therefore, appropriate content of EPA and DHA in diets may enhance larval growth, though no statistical relationship between the two fatty acids with growth parameters was observed. This could be explained in that little data about EPA and DHA in the experimental oil sources were available for analysis, except fish oil. It was demonstrated that black soldier flies can hardly biosynthesize C20 poly-unsaturated fatty acid in vivo [36]. Nonetheless, early studies on *Drosophila* melanogaster indicated that PUFA, especially EPA and DHA, may have decisive influences on the normal growth and development of larvae, and play essential roles in the immune response to stressors as crucial signaling molecules [37,38]. The role of EPA and DHA in black soldier flies may have been neglected before, since the two fatty acids are deficient in most of the rearing substrates.

The influence of dietary fatty acids on insect development differs between species. For example, a diet containing palm oil significantly shortened the developmental time, reduced mortality of young nymphs, and increased the female fertility rate of *Arma chinensis* [39]. The addition of wheat germ oil rich in linolenic acid to the fruit fly rearing diet was effective in improving fruit fly quality, especially in egg hatch, fliers, egg production, and pupal recovery [40]. In this study, the pre-pupa rate of insects was statistically influenced by dietary content of C18:0, C18:3n-3 and C18:2n-6, which were responsible for 92.7%, 30.1% and −25.8% of the variation in larval pre-pupa rate, respectively. Dietary 18:0 was reported to regulate mitochondrial morphology and play crucial roles for energy support in Drosophila through a dedicated signaling pathway [41]. C18:3n-3 and C18:2n-6 were commonly considered direct precursors of the higher poly-unsaturated acids in insects, and appropriate dietary levels of 18:3n-3 significantly boosted the pre-pupa rate and emergence rate of some insects [42]. Regarding C18:2n-6, black solider fly larvae bioaccumulated only around 13% of this fatty acid and metabolized approximately two-thirds of it into saturated fatty acids, as suggested by a previous study. The possible reason was that an excessive amount of C18:2n-6 might damage black soldier fly development [36]. Similarly, dietary C18:2n-6 over 0.1% reduced the pupation percentage of *Apis mellifera ligustica* [43], and C18:3n-3 was more effective than C18:2n-6 in promoting emergence in the pink bollworm *Pectittophora gossypielld* [44]. Therefore, further evidence is required in the future to confirm the effects of the three fatty acids by measuring more parameters related to larval development.

### 4.2. The Nutrient Composition of Larvae

In the present study, the inclusion of dietary fat increasing from 5% to 10% led to a significant increase of fat levels (15.0% to 23.8%) in larvae. Similarity, a fat increase was also observed in larvae fed with higher fat/energy rearing substrates compared with those fed with higher protein or lower energy substrates [21,45,46].

Larval fat content was significantly affected by dietary fat sources as well. The highest fat content was achieved in larvae fed with the 10% lard oil, coconut oil and fish oil, which were all rich in SFA (35.9% to 84.8%). This could be explained by a higher synthesis of fatty acids by utilizing dietary fatty acids, mainly saturated fatty acids. It was suggested that black solider fly larvae could synthesize acetyl-CoA by fatty acids C12:0, C14:0, C16:0, and C18:0, and finally formed triglycerides [47]. Meanwhile, the contribution of these fatty acids to tissue triglyceride biosynthesis was crucial [36]. Protein content in larvae varied conversely with that of fat. Wang et al. [48] reported that the sum of body protein and fat content maintained at a constant level of 75.9 ± 2.6% in 86 species of insects, regardless of their food types. Concerning the black soldier fly, larval fat content was strongly affected by dietary nutrient concentration, while larval protein content in the black soldier fly varied within narrow limits [49]. Additionally, larval final body weight and size differed greatly between groups at the sampling time in this study. Those who had bigger weight and size may accumulate more fat and less protein in their body, as demonstrated in other studies [49,50,51]. Nevertheless, protein synthesis is a quite complex process involving a wide range of nutrients, hormones and intestinal microorganisms [52]. This should be further investigated.

### 4.3. Fatty Acid Composition

By comparing the relationships of the fatty acids between the final larvae and diets, most of the fatty acids could be grossly divided into two categories:(1)C12:0, C14:0 and C16:1n-9. Their proportions in larvae were much higher in larvae than those in diets. For example, C12:0 was the most abundant (17.4–44.9%) fatty acid in larvae in all groups, but it can hardly be found in most of the oil sources except that of coconut oil. Similarly, a larger proportion of C14:0 (3.9–15.2%) and C16:1n-9 (1.3–3.7%) in larvae over most diets was determined, compared with that of them in diets. According to previous studies, the high proportion of C12:0 in larvae can distinguish the black soldier fly from other insects [17,53,54]. Several saturated fatty acids including C10:0, C12:0, and C14:0 were biosynthesized through the elongation process with carbohydrate as an essential source of acetyl-CoA. In addition, C16:1n-9 was produced through a desaturation process based on 16:0 [36]. Our data seemed to support this suggestion. However, C10:0 was not determined in larvae in this study. Moreover, it was apparent that dietary fatty acids and carbohydrates contributed to the bioaccumulation of C12:0 and C14:0. They were properly bioaccumulated by larvae up to almost 44.9% and 15.2% in the coconut group, respectively, indicating that the larvae may incorporate C12:0 and C14:0 when they occurr in diets. Interestingly, higher dietary fat content could enhance the bioaccumulation of C12:0 and C14:0. Hence, the enrichment of the nutritive value of C12:0 in black soldier fly larvae by manipulating fat content and fatty acid profiles may be technically feasible.(2)C16:0, C18:1n-9, C18:2n-6 and C18:3n-3. Their proportions in larvae were linearly related to those of them in diets. Compared with previous studies, the present study presented a quantitative assessment of the relationship of these fatty acids between larvae and diets, so that the proportions of the four fatty acids in larvae could be estimated from their proportions in diets. According to Hoc et al. (2020) [36], 16:0 and 18:1n-9 thought to be merely bioaccumulated from diets were also produced by the larvae itself, 18:2n-6 and 18:3n-3 could be partly retained in larval body, and the rest of them could be generated into acetyl-CoA through fatty acid β-oxidation. Our results were partly consistent with these suggestions. Specifically, C18:2n-6 and C18:3n-3 proportions in larvae (5.5–26.1% and 0.5–23.2%) were significantly reduced compared to those in diets (6.5–49.8% and 0.9–50.0%), reflecting that it was less economic for larvae to acquire the two fatty acids from diets. However, the larger proportions of C16:0 in larvae (10.3–15.5%) over diets (6.7–11.0%) were observed in linseed oil, peanut oil, coconut oil and soybean oil groups, while its proportions in larvae were around 20% in the lard oil and fish oil groups, despite the two diets containing high proportions of 16:0 (21.9–25.9%). Similar results were revealed for 18:1n-9. Therefore, we expected that the bioaccumulation of the two fatty acids in the body was limited, and that the extra part of them may be used for supplying oxidation energy, or biosynthesis as other fatty acids. Therefore, despite the high levels of C18:2n-6 and C18:3n-3 in plant oil, and EPA and DHA in fish oil, possibilities to tailor them into black soldier fly larvae through diets maybe limited and not cost-effective. Nevertheless, as a promising source of SFA, especially C12:0, black soldier fly larvae could work well in combination with oil rich in MUFA and PUFA to satisfy the fatty acid requirement of aquatic animals.

## 5. Conclusions

In this study, two levels of six dietary fat sources were fed to black solider fly larvae. The influence of dietary fatty acids was quantitatively determined by measuring parameters related to larval growth, development and nutrient composition. It can be concluded that (1) the pre-pupa rate of larvae could be influenced by dietary C18:0, C18:3n-3 and C18:2n-6 content, while larval final body weight was affected by that of C16:0 content. (2) Several fatty acids (C16:0, C18:1n-9, C18:2n-6 and C18:3n-3) could be partially incorporated from diets and retained in larvae bodies. Therefore, possibilities to tailor PUFA into black soldier fly larvae through the diets may be limited. However, dietary PUFA may play specific roles in enhancing larval growth. (3) Larvae could produce C12:0, C14:0, C16:1n-9 in vivo when diets were free of them. The quantitative assessment presented some nutrient strategies for the better use of black soldier fly larvae as feed sources, for example, the inclusion of an appropriate amount of black soldier fly larvae in aquafeed as the main source of SFA and lauric acid, as well as a partial source of protein, MUFA and PUFA to save fish meal and fish oil.

Currently, the fat metabolism process of the black soldier fly has attracted wide attention for its purposeful application as promising oil sources, such as dietary oil, edible oil, or biodiesel. Therefore, the specificity and complexity of carboxylation, catabolism, desaturation and isomerization processes involved in fat metabolism in this species should be further investigated, and various formulated species-specific diets for fulfilling nutritional needs should be explored in the future. The present study provided a preliminary observation of the impact of dietary fatty acid on larval growth and fatty acid profile. The limitation of the trial is that some factors which may influence the accuracy of the quantitative assessment could not be taken into account. For example, the varying protein and energy level with the changed fat level of the experimental diets may increase the complexity of the fatty acid requirement assessment, and micronutrients may have played roles in fatty acid metabolism. Therefore, a standard and precise feed formula for larval nutrition research is needed, and research of micronutrients on larval growth and metabolism may be valued for efficient larval culture and quality improvement.

## Figures and Tables

**Figure 1 animals-12-00486-f001:**
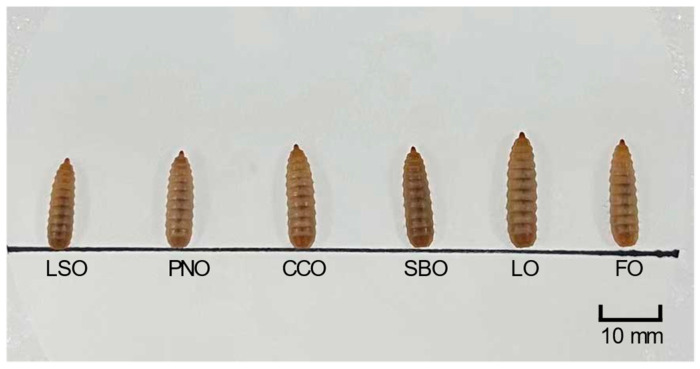
Size differences between larvae fed with six fat sources (5% supplemental level) on the 19th day. LSO: linseed oil; PNO: peanut oil; CCO: coconut oil; SBO: soybean oil; LO: lard oil; FO: fish oil.

**Figure 2 animals-12-00486-f002:**
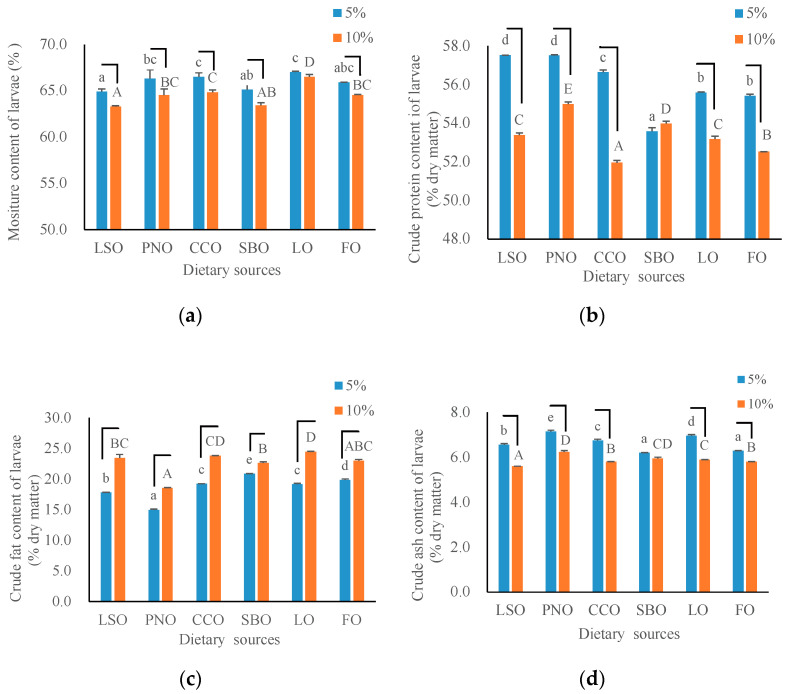
Nutrient composition of larvae fed with diets containing two levels of six fat sources. LSO: linseed oil; PNO: peanut oil; CCO: coconut oil; SBO: soybean oil; LO: lard oil; FO: fish oil. (**a**–**d**) Represents moisture, crude protein, crude fat and crude ash content of larvae affected by dietary sources and levels. Each column represents the mean value of four replicates. The lower case letters above the blue column mean significant differences existing between groups at 5% supplemental fat level, and the capital letters above the red column mean significant differences existing between groups at 10% supplemental fat level (One-way ANOVA and Tukey’s test, *p* < 0.05). The fold line across the blue and red column means values with significant differences between 5% and 10% supplemental fat levels (One-way ANOVA, *p* < 0.05).

**Figure 3 animals-12-00486-f003:**
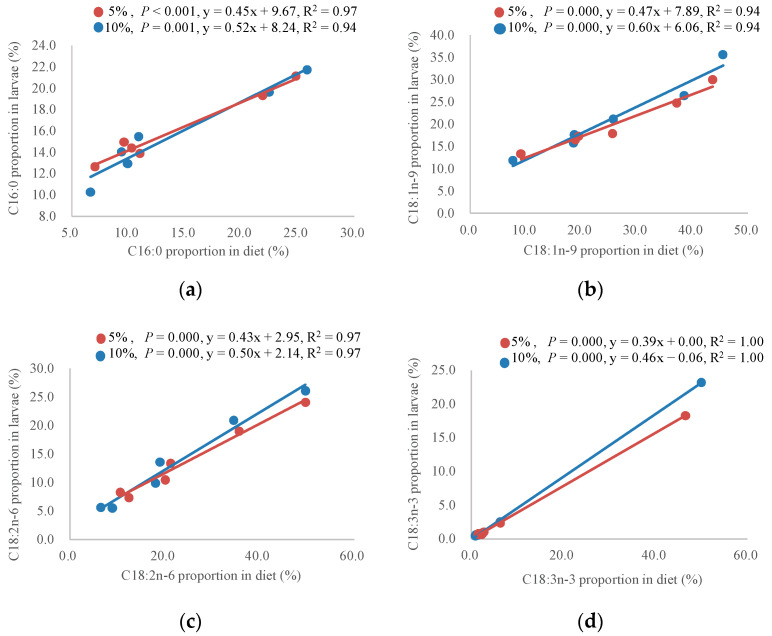
Graphical representation of the fatty acid proportions in the larvae in relation to those in diets. (**a**) C16:0 proportions in the larvae in relation to those in diets. (**b**) C18:1n-9 proportions in the larvae in relation to those in diets. (**c**) C18:2n-6 proportions in the larvae in relation to those in diets. (**d**) C18:3n-3 proportions in the larvae in relation to those in diets. The linear regression equations are presented in each panel, accompanied with the fitting lines. *p*-Values  <  0.05 mean a significant influence of dietary fatty acid proportions on larval fatty acid proportions.

**Table 1 animals-12-00486-t001:** The formulation and nutrient composition of the experimental diets containing two levels of six fat sources.

Ingredients	Content (%)											
Soybean meal	95	90	95	90	95	90	95	90	95	90	95	90
Linseed oil	5	10										
Peanut oil			5	10								
Coconut oil					5	10						
Soybean oil							5	10				
Lard oil									5	10		
Fish oil											5	10
Nutrient composition of diets (dry weight)												
Crude protein (%)	44.1	41.1	43.7	41.1	43.5	42.1	44	41.5	43.5	41.5	44.3	41.4
Crude lipid (%)	5.8	9.8	5.6	9.9	6.3	10.3	6.1	10.3	5.6	9.7	6.1	10.0
Crude ash (%)	5.9	5.5	5.8	5.6	5.9	5.6	5.9	5.6	5.9	5.7	5.8	5.6
Gross energy (KJ/kg)	18.7	19.6	18.1	19.6	18.8	19.9	18.8	19.6	18.1	19.0	18.8	19.6
Lysine (%)	2.50	2.35	2.52	2.32	2.52	2.38	2.52	2.40	2.48	2.32	2.48	2.39
Methionine (%)	0.55	0.52	0.54	0.53	0.55	0.51	0.58	0.54	0.52	0.50	0.56	0.49
Arginine (%)	2.89	2.82	2.88	2.85	2.85	2.85	2.91	2.85	2.88	2.82	2.90	2.82
Ca (%)	0.32	0.30	0.31	0.29	0.33	0.30	0.30	0.28	0.32	0.29	0.31	0.28
P (%)	0.60	0.56	0.58	0.56	0.60	0.57	0.59	0.56	0.59	0.56	0.58	0.56
Cl (%)	0.08	0.07	0.07	0.08	0.07	0.08	0.07	0.05	0.13	0.08	0.08	0.07
Na (mg/kg)	140	130	110	89	140	120	110	97	97	100	110	98

**Table 2 animals-12-00486-t002:** Fatty acid composition (expressed as percentage of total fatty acids) of the experimental diets containing two levels of six fat sources.

Dietary Sources and Levels		C8:0	C10:0	C12:0	C14:0	C16:0	C16:1n-7	C18:0	C18:1n-9	C18:2n-6	C18:3n-3	C20:1n-9	C20:5n-3	C22:6n-3	Others	SFA	MUFA	PUFA
Linseed oil	5%	-	-	-	0.1	7.0	0.1	4.5	19.5	21.3	46.5	0.1	-	-	0.9	12.1	19.8	68.1
	10%	-	-	-	0.1	6.7	0.1	4.6	18.7	19.0	50.0	0.1	-	-	0.8	11.7	19.0	69.3
Peanut oil	5%	-	-	-	0.1	9.6	0.1	3.7	43.6	35.8	1.5	1.0	-	-	4.7	17.8	45.0	37.2
	10%	-	-	-	0.1	9.4	0.1	3.6	45.5	34.6	0.9	1.1	-	-	4.8	17.7	46.9	35.4
Coconut oil	5%	4.8	4.5	38.6	16.0	10.3	-	3.9	9.1	10.6	1.6	0.1	-	-	0.6	78.7	9.2	12.2
	10%	5.4	5.0	42.7	17.5	9.9	-	3.8	7.7	6.5	0.9	0.1	-	-	0.5	84.8	7.8	7.4
Soybean oil	5%	-	-	-	0.1	11.0	0.1	5.2	25.6	49.8	6.3	0.2	-	-	1.6	17.9	26.0	56.1
	10%	0.1	0.1	-	0.1	10.9	0.1	5.3	25.7	49.8	6.2	0.2	-	-	1.5	17.9	26.1	56.0
Lard oil	5%	-	-	0.1	1.0	24.9	1.7	11.0	37.2	20.2	2.2	0.5	-	-	1.4	37.5	39.5	23.0
	10%	-	-	0.1	1.0	25.9	1.8	11.4	38.5	18.1	1.3	0.5	-	-	1.3	39.0	41.0	20.1
Fish oil	5%	-	-	0.1	5.5	21.9	5.9	5.3	18.9	12.5	2.7	2.4	7.4	10.7	6.6	35.9	28.9	35.2
	10%	-	-	0.1	6.0	22.5	6.4	5.2	18.6	9.0	2.3	2.6	8.3	12.1	7.1	37.1	29.4	33.5

“-” indicated that the fatty acid content was below the detection limit. SFA, MUFA and PUFA indicated saturated fatty acid, mono-unsaturated fatty acid and poly-unsaturated fatty acid, respectively.

**Table 3 animals-12-00486-t003:** Growth parameters of black soldier fly larvae fed with diets containing two levels of six fat sources.

Growth Parameter	Fat Levels	Fat Sources					
		Linseed Oil	Peanut Oil	Coconut Oil	Soybean Oil	Lard Oil	Fish Oil
Pre-pupa rate ^1^ (%)	5%	0.0 ± 0.0 ^aA^	0.0 ± 0.0 ^a^	0.0 ± 0.0 ^aA^	1.1 ± 0.7 ^a^	7.7 ± 0.3 ^bA^	3.2 ± 1.2 ^ab^
	10%	11.8 ± 1.2 ^dB^	0.0 ± 0.0 ^a^	5.7 ± 0.8 ^cB^	1.2 ± 1.2 ^b^	19.8 ± 1.6 ^eB^	6.2 ± 1.5 ^c^
Final body weight (mg/individual)	5%	59.5 ± 10.7 ^aA^	55.9 ± 7.8 ^aA^	103.7 ± 11.4 ^bA^	104.3 ± 7.3 ^b^	118.7 ± 13.8 ^c^	113.7 ± 17.5 ^cA^
	10%	71.4 ± 6.2 ^aB^	94.2 ± 9.5 ^bB^	115.5 ± 8.7 ^cB^	100.0 ± 9.7 ^b^	115.6 ± 14.4 ^c^	147.2 ± 13.9 ^dB^
Final body size (mm/individual)	5%	13.3 ± 1.1 ^b^	12.2 ± 0.7 ^a^	15.3 ± 0.7 ^bc^	15.3 ± 0.5 ^bc^	15.8 ± 0.7 ^cd^	16.0 ± 0.8 ^d^
	10%	13.0 ± 0.8 ^a^	15.5 ± 0.7 ^bc^	15.8 ± 0.8 ^c^	15.1 ± 0.6 ^b^	15.9 ± 0.8 ^c^	17.1 ± 0.7 ^d^

Descriptive statistics are presented as means of four replicates ± the standard deviation. The subscript lower case letters in the same line mean significant differences existing between the values in the same line (fat sources) (One-way ANOVA and Tukey’s test, *p* < 0.05), and the subscript capital letters in the same column mean significant differences existing between values in the same column (fat levels) (One-way ANOVA, *p* < 0.05). ^1^ Pre-pupa rate = 100% × (number of pre-pupae/total number of the alive individuals).

**Table 4 animals-12-00486-t004:** Multiple stepwise regression analysis of fatty acid content in diets and larval pre-pupa rate and final body weight.

Models	R^2^	*p*-Value
Pre-pupa rate = 0.879 × C18:0 content	0.749	<0.001
Pre-pupa rate = 0.908 × C18:0 content + 0.298 × C18:3n-3 content	0.829	<0.001
Pre-pupa rate = 0.927 × C18:0 content + 0.301 × C18:3n-3 content − 0.258 × C18:2n-6 content	0.899	<0.001
Final body weight = 0.758 × C16:0 content	0.531	0.004

R^2^ is the coefficient of determination, which is used for the evaluation of goodness-of-fit of the fitting model. *p*-Values <  0.05 mean a significant influence of dietary fatty acid content on larval pre-pupa rate or final body weight.

**Table 5 animals-12-00486-t005:** Fatty acid composition (expressed as percentage of total fatty acids) of the larvae reared on the experimental diets containing two levels of six fat sources.

Fatty Acids		C12:0	C14:0	C16:0	C16:1n-9	C18:0	C18:1n-9	C18:2n-6	C18:3n-3	C20:5n-3	Others	SFA	MUFA	PUFA
Initial larvae		0.7 ^a^	0.5 ^a^	12.9 ^bc^	2.2 ^c^	13.5 ^h^	29.5 ^h^	33.8 ^k^	3.3 ^d^	0.1 ^a^	3.4 ^f^	30.0 ^a^	32.5 ^h^	37.5 ^j^
Linseed oil	5%	23.9 ^e^	5.6 ^e^	12.6 ^b^	2.6 ^c^	4.5 ^ef^	17.3 ^d^	13.4 ^f^	18.2 ^e^	-	1.9 ^b^	47.6 ^e^	20.5 ^c^	31.9 ^i^
	10%	23.8 ^e^	4.9 ^d^	10.3 ^a^	1.3 ^a^	3.6 ^cd^	17.7 ^d^	13.6 ^f^	23.2 ^f^	0.1 ^a^	1.6 ^a^	43.6 ^c^	19.3 ^b^	37.1 ^j^
Peanut oil	5%	21.0 ^c^	4.8 ^c^	15.0 ^g^	2.5 ^c^	4.4 ^e^	30.0 ^h^	19.0 ^g^	0.7 ^ab^	-	2.7 ^e^	46.9 ^de^	33.4 ^i^	19.7 ^e^
	10%	17.4 ^b^	3.9 ^b^	14.0 ^ef^	1.5 ^a^	3.7 ^cd^	35.7 ^i^	20.9 ^h^	0.5 ^a^	-	2.5 ^de^	40.7 ^b^	37.9 ^j^	21.3 ^f^
Coconut oil	5%	38.7 ^i^	14.0 ^i^	14.4 ^ef^	4.3 ^e^	3.5 ^c^	13.4 ^b^	8.3 ^c^	0.8 ^ab^	-	2.6 ^de^	72.2 ^j^	18.7 ^b^	9.1 ^b^
	10%	44.9 ^j^	15.2 ^j^	13.2 ^cd^	3.7 ^d^	2.7 ^a^	11.9 ^a^	5.6 ^a^	0.6 ^a^	-	2.5 ^d^	77.3 ^k^	16.5 ^a^	6.2 ^a^
Soybean oil	5%	28.5 ^h^	5.8 ^f^	13.9 ^de^	1.9 ^b^	3.7 ^cd^	17.9 ^d^	24.1 ^i^	2.4 ^c^	0.1 ^a^	1.8 ^b^	53.1 ^f^	20.4 ^c^	26.5 ^g^
	10%	22.2 ^d^	4.9 ^d^	15.5 ^h^	1.9 ^b^	3.9 ^d^	21.2 ^e^	26.1 ^j^	2.5 ^c^	0.4 ^b^	1.5 ^a^	47.5 ^e^	23.6 ^d^	28.9 ^h^
Lard oil	5%	25.8 ^f^	5.9 ^f^	21.1 ^j^	3.7 ^d^	5.4 ^g^	24.8 ^f^	10.5 ^e^	0.7 ^ab^	-	2.3 ^c^	59.3 ^h^	29.2 ^f^	11.4 ^c^
	10%	25.5 ^f^	5.6 ^e^	21.7 ^j^	3.6 ^d^	4.6 ^f^	26.5 ^g^	9.9 ^d^	0.6 ^a^	0.1 ^a^	2.1 ^c^	58.4 ^g^	30.7 ^g^	10.9 ^c^
Fish oil	5%	27.4 ^g^	7.7 ^g^	19.3 ^i^	6.0 ^f^	3.7 ^cd^	16.6 ^c^	7.3 ^b^	1.0 ^b^	4.8 ^c^	6.2 ^g^	60.5 ^hi^	24.2 ^e^	15.3 ^d^
	10%	27.6 ^gh^	8.0 ^h^	19.6 ^i^	6.7 ^g^	3.2 ^b^	15.8 ^c^	5.5 ^a^	0.9 ^ab^	5.6 ^d^	7.1 ^h^	61.1 ^i^	24.4 ^e^	14.5 ^d^

“-” indicated that the fatty acid content was below the detection limit. The subscript lower case letters in the same column mean significant differences existing between the values in the same column (One-way ANOVA and Tukey’s test, *p* < 0.05). SFA, MUFA and PUFA indicated saturated fatty acid, mono-unsaturated fatty acid and poly-unsaturated fatty acid, respectively.

## Data Availability

Raw data are not publicly available.

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
