# Peer review of "Growth and Fatty Acid Composition of Black Soldier Fly Hermetia illucens (Diptera: Stratiomyidae) Larvae Are Influenced by Dietary Fat Sources and Levels"

_animals, 2022, doi:10.3390/ani12040486_

Round 1
Reviewer 1 Report
The authors investigated the effect of dietary fat sources and levels on growth performance and fatty acid composition of black soldier fly (Hermetia illucens) larvae. They designed twelve treatments to test the best lipid source. This manuscript (MS) was clearly written and easy to understand. This work can help the sustainability of aquaculture as black soldier fly has been introduced as one of the best ingredients for fish meal replacement. However, some minor issues significantly compromised the quality of this MS.
Minor comments
Abstract
- Line 22, please revise it, “will” is a too strong verb.
- Line 28 and throughout the MS, please report the fatty acids with “C”. For example, C18:0, C18:3n-3..
- Line 30, what is the point of this sentence? I mean, reporting the range does not say anything.
- Line 29, 53% is not strong enough; please delete it.
- Line 31, is not surprising as the contents of some fatty acids in more in the animal body than their diet due to fatty acids biosynthesis. And how much is “Much more proportions”. Please write it numerically. Please revise these sentences.
- Line 34, it is not clear based on which information you suggest this; please revise it.
- Line 37, the main output of this study can be which one is closer to the fish oil profile. Please add it to the abstract and discussion section.
Introduction:
- Well-developed introduction and included a clear fellow and relevant points.
- Line 41- 50, please revise this section and focus on fish and aquaculture.
Material and methods
- Well-organized section. Clear fellow and all required details were provided.
- Line 86-88, please check the error in reference.
- Please provide the complete experimental diets formulation or the feeds they ate. Providing only the approximate nutrient composition is not enough. Further, please provide energy content as well. You can have both formulation and approximate composition in a table.
- Line 120, please make sure you defined all abbreviations in the MS for the first time.
- Table 2, it is surprising how adding 5% or 10% oil to diets can hugely change the fatty acids profile. Further, there are few changes between 5% and 10%. Providing the diet formulation can illustrate this issue better. By the way, please recheck the fatty acids profile. The sum of fatty acids should usually be less than 95%.
Results
- Well-written section, all necessary things have been covered.
- Line 153, check ref
- Line 65, what was the control diet? Please add it to table 1 diet formulation.
- Line 167, fix ref
- Line 174, please add it in which treatment growth was higher; I could not find it in the text.
- Table 3, please tidy this table up. Separate the P values in a column or raw. You can simply state that if the P-value was lower than 0.05, you report the subsets and no need to report all P values. Further, if there is no significant difference no need to report subsets.
- Table 2, what is the unit of fatty acids?. Please provide it.
- Line 197, fix the error.
- Table 4, what does “content” stand for?
- Line 203, chemical composition?
- Line 232-233, fix error in ref here and throughout the discussion.
Discussion
- Very well prepared, I have no comment!!
- Please read this part and others to check for fixing some language errors.
- Please suggest future research regarding the application of this output in aquaculture.
- Please highlight that FO was not better than other oils for larvae, and it is a great output for aquaculture.
Best regards
Reviewer 2 Report
Diet composition should be supplied in terms of proximate composition and at least the major amino acids. It should also include basic minerals such as Ca and P while Na and Cl would add value. Energy is also a major component and totally neglected here. Lipid as such adds a tremendous amount of energy and the authors do not acknowledge or discuss that. Further the digestibility / availability of fats are dependent on the level of saturation - 5% lard is not the same as 5% soya oil. Thus different levels of energy was supplied even at similar fat levels. Growth and development is first and foremost influenced by energy content of the feed. The insect industry is way beyond such basic formulation as is used in this trial.
Technical preparation is insufficient, links are not working so in text references are not linked to relevant tables or figures.
Grammatical errors are present all over the article - to a level where the meaning is lost. Extensive language editing is needed.
A great concern for me are statements like "the insects could survive on a lipid free diet" - it seems like the authors do not understand nutrient composition of the composition of the raw materials which they worked with. Soyabean meal inherently contains lipids. When looking at the table for the composition of the feed it does not make sense. Considering a 2.5% fat content for the soyabean meal and then the addition of lipid on top of that that the fat content should exceed that of the addition or at least vary in the same proportions. It does not make sense that the 5% diet has in excess of 5% lipid while the 10% diet has less than 10% lipid.
Another concern is the final moisture content of the feed - if they have soyabean meal with 10% moisture (should be between 10 and 15%) and they add 50% water then the final moisture content will not be 70%?
The discussion and conclusion do not address all factors - essentially the trial is confounded and the limitations not acknowledged.
Reviewer 3 Report
A good paper. This study is well designed and the data generated presents new information. However, some changes are suggested to improve clarity and presentation.
Although the text is comprehensible, somewhat excessive number of minor errors in grammar and spelling are found. Some errors are listed below, as examples – but careful proof-reading is recommended.
L15: Recent research has ..
L16:..are not comparable
L18: delete ‘quite’
L20: insert the zoological name after ‘fly’; also in L44
L23: orientation culture?? Not clear
L30: different unit is used
L34: ..varying ..
L41: Research on the …..
L55: …vary..
L63: present data ?? not clear
L69: delete ‘quite’
L70: ..fatty acid composition, not compositions – change here and elsewhere
L70: ..to rear ..
L71: delete ‘major’
L76: purposive applications?? Not clear.
The above are only examples. Suggest proof-reading by an Editor
OTHER ISSUES
TITLE: Growth and fatty acid composition …….. larvae are influenced ….; Also change ‘growth performance’ to ‘growth’ throughout the manuscript
Just an observation: do you not need an ethics approval for insect studies
L82-3: not clear. rephrase. The oils were added on top of soybean meal or replaced (w/w) the meal ?
L98: clearly indicate that the trial lasted for 16d from d3 to d19.
L104: using 20 larvae for size is understandable – but could have used more for weight. How exactly these measurements were made?. Expand.
Table 3: Please confirm that the initial weights of treatment groups were not significantly different and indicate in the table or as a footnote. I believe that weight gain is more relevant than final weight. A comment in the text is instructive.
Figure 1 legend: Size differences between …… level) on d19
L202: Nutrient composition … (the term approximate is incorrect). Change throughout.
Table 5: row 1: 12:0, not 12:00. Change all.
Reviewer 4 Report
The study evaluates the growth performance parameters and nutrient compositions of larvae of black soldier fly fed with different dietary fat levels, since the larvae are promising and sustainable feed sources. The study provides a better understanding and new knowledge for its applications. However, part of the manuscript lack clarity, has numerous grammatical mistakes, formatting errors, and need to be proofread for consistency and clarity. Many sentences appear to have missing citations. There are excessive use of the word "quite", please check the entire manuscript. My specific and additional comments can be found in the file attached.
